# Exploring Folklore Ecuadorian Medicinal Plants and Their Bioactive Components Focusing on Antidiabetic Potential: An Overview

**DOI:** 10.3390/plants13111436

**Published:** 2024-05-22

**Authors:** Soham Bhattacharya, Neha Gupta, Adéla Flekalová, Salomé Gordillo-Alarcón, Viviana Espinel-Jara, Eloy Fernández-Cusimamani

**Affiliations:** 1Department of Agroecology and Crop Production, Faculty of Agrobiology, Food and Natural Resources, Czech University of Life Sciences Prague, Kamýcká 129, Prague 6, 16500 Suchdol, Czech Republic; bhattacharya@af.czu.cz; 2Department of Crop Sciences and Agroforestry, Faculty of Tropical AgriSciences, Czech University of Life Sciences Prague, Kamýcká 129, Prague 6, 16500 Suchdol, Czech Republic; guptan@ftz.czu.cz (N.G.); xflea003@studenti.czu.cz (A.F.); 3Department of Medicine, Faculty of Health Sciences, Universidad Técnica del Norte, Avda. 17 de Julio 5-21, Ibarra 100105, Ecuador; asgordillo@utn.edu.ec; 4Department of Nursing, Faculty of Health Sciences, Universidad Técnica del Norte, Avda. 17 de Julio 5-21, Ibarra 100105, Ecuador; vmespinel@utn.edu.ec

**Keywords:** antidiabetic, antihyperglycemic, bio-active compound, hypoglycemic, medicinal plant

## Abstract

Diabetes mellitus (DM) is a global health concern characterized by a deficiency in insulin production. Considering the systemic toxicity and limited efficacy associated with current antidiabetic medications, there is the utmost need for natural, plant-based alternatives. Herbal medicines have experienced exponential growth in popularity globally in recent years for their natural origins and minimal side effects. Ecuador has a rich cultural history in ethnobotany that plays a crucial role in its people’s lives. This study identifies 27 Ecuadorian medicinal plants that are traditionally used for diabetes treatment and are prepared through infusion, decoction, or juice, or are ingested in their raw forms. Among them, 22 plants have demonstrated hypoglycemic or anti-hyperglycemic properties that are rich with bioactive phytochemicals, which was confirmed in several in vitro and in vivo studies. However, *Bryophyllum gastonis-bonnieri*, *Costus villosissimus*, *Juglans neotropica*, *Pithecellobium excelsum*, and *Myroxylon peruiferum*, which were extensively used in traditional medicine preparation in Ecuador for many decades to treat diabetes, are lacking in pharmacological elucidation. The Ecuadorian medicinal plants used to treat diabetes have been found to have several bioactive compounds such as flavonoids, phenolics, fatty acids, aldehydes, and terpenoids that are mainly responsible for reducing blood sugar levels and oxidative stress, regulating intestinal function, improving insulin resistance, inhibiting α-amylase and α-glucosidase, lowering gluconeogenic enzymes, stimulating glucose uptake mechanisms, and playing an important role in glucose and lipid metabolism. However, there is a substantial lack of integrated approaches between the existing ethnomedicinal practices and pharmacological research. Therefore, this review aims to discuss and explore the traditional medicinal plants used in Ecuador for treating DM and their bioactive phytochemicals, which are mainly responsible for their antidiabetic properties. We believe that the use of Ecuadorian herbal medicine in a scientifically sound way can substantially benefit the local economy and industries seeking natural products.

## 1. Introduction

Diabetes mellitus (DM) represents a major public health challenge triggered by rising obesity rates and characterized by high comorbidity, reduced quality of life, and premature mortality, as well as major economic and societal expenses [1]. Long-term diabetes can cause a variety of complications, including microvascular complications like nephropathy, neuropathy, and retinopathy, as well as macrovascular complications like atherosclerosis, heart attack, and peripheral blood vessel disease [2,3]. The pervasiveness of diabetes has increased at the national, regional, and global levels [4]. Additionally, the International Diabetes Federation (IDF) anticipates that 578 million individuals are expected to have DM by 2030, escalating to 629 million by 2045, with a global prevalence of 9.9% [5]. Furthermore, the average prevalence of diabetes in South and Central America reached 9.4% in adults, impacting around 32 million people. Ecuador additionally ranked among the 20 countries with the highest prevalence of DM (5.5%) during that period [6]. Following the IDF 2021 study, 4.7% of Ecuadorians aged 20–79 had diabetes [7]. Diabetes was ranked as the second-largest cause of mortality in Ecuador in 2021 after COVID-19 [8]. Diabetes-related expenditures per person are estimated at approximately 2280 USD annually [7]. Dismantled health systems and high out-of-pocket health costs impose an enormous financial burden on individuals for effective disease management [9]. Diabetes is an intricate condition that renders individuals susceptible to various repercussions such as cardiovascular disease, chronic kidney disease (CKD), cancer, TB, influenza, and COVID-19, thereby necessitating multiple therapeutic approaches [10,11]. Diabetes therapy includes insulin and oral antidiabetic medications such as sulfonylurea, biguanides, α-glucosidase inhibitors, and glinides. Nonetheless, in nations with limited resources, these medications are expensive and challenging to obtain [12]. These treatments have their own set of limitations, ranging from the emergence of resistance and undesirable side effects to a lack of sensitivity in vast sectors of the patient population. These treatments are associated with adverse reactions; for example, thiazolidinediones prompt liver toxicity; sulphonylureas could exacerbate heart disease, reduce blood glucose levels below the normal range, and increase body weight gain; and bloating, flatulence, diarrhea, and abdominal discomfort and pain are the most frequently reported issues with glucosidase inhibitors [13,14]. The shortcomings of the current existing oral antidiabetic medications, particularly in terms of efficacy and safety, combined with the disease’s growth as a global epidemic, have prompted a coordinated effort to discover drugs that may more effectively treat diabetes [15]. Furthermore, with a growing prevalence of DM in rural communities around the world, as well as the detrimental effects of synthetic medicine, there is an imperative need for the development of indigenous, inexpensive botanical sources for antidiabetic crude or purified drugs [16,17].

During the last few decades, several medicinal plants like *Allium sativa* L., *Trigonella Graecum Foenum*, *Silybum marianus*, *Citrullus colocynthis*, *Zingiber officinale*, etc., have been useful in treating diabetes and are used empirically as antidiabetic and antihyperlipidemic remedies [18,19,20,21,22]. Ecuadorian herbs like *Foeniculum vulgare*, *Cajanus cajan*, *Urtica dioica*, *Verbena litoralis*, etc., used in traditional medicine were found to exhibit antidiabetic and antihypoglycemic qualities [23,24,25,26,27]. Furthermore, the amalgamation of bioactive components present in the above-mentioned plants is not only responsible for combating diabetes but also for antimicrobial, anti-inflammatory, antioxidant, and antifungal activities [28,29]. For example, bioactive compounds like tau-cadinol, α-cadinol, spathulenol, caryophyllene oxide, trans-anethole, fenchone, estragole, limonene, α-phellandrene, etc., impart the antidiabetic property to the Ecuadorian herbs and have demonstrated positive results during clinical trials [28,29,30,31].

Several Ecuadorian ethnic groups rely extensively on medicinal plants to safeguard their medical conditions [32]. The clinical expertise of traditional healer practitioners has been meticulously maintained over thousands of years [23]. Therefore, herbal drugs have gained legitimacy throughout the years because of the apparent efficacy and safety of the plants [33]. As a result, many doctors, particularly those in intercultural, government-supervised health districts, now practice a combination of contemporary and conventional medicines. Much ethnopharmacological research has been carried out in Ecuador. Despite being a multi-cultural country, reviews of ethnopharmacological studies, particularly for the treatment of diabetes, are scarce. This review provides a systematic analysis of recent phytochemical and antidiabetic studies conducted on Ecuadorian medicinal plants, with a particular emphasis on the potential relationships between traditional applications and pharmaceutical properties, as well as the therapeutic potential of natural remedies. Plant species can consist of both native and introduced species. Thus, we incorporated pharmacological evidence (preclinical and clinical trials), phytochemical concerns, and toxicological assessments. The results of our reviews are expected to highlight gaps within existing research and provide critical direction for future frameworks aimed at developing cost-effective and novel phytomedicines that combat DM.

## 2. Materials and Methods

The methodology of this review was based on the scientific information gathered mainly from available online scientific publications from the last 20 years (2003–2024) in the English and Spanish languages to obtain the most recent data. Databases like Scopus, ScienceDirect, ResearchGate, SciELO, PubMed, and Web of Science were searched based on keywords such as “traditional medicine of Ecuador”, “medicinal plants with antidiabetic activity”, “ethnopharmacological activities”, “plants used for the treatment of diabetes”, “ethnobotanical surveys of Ecuador”, “phytochemical composition of medicinal plants”, “chemical constituents”, “secondary metabolite”, and “phytochemical”. We also used a published book to gather knowledge about Ecuadorian plants used in traditional medicine [34]. 

The selection of plant species with antidiabetic potential from Ecuador and their bioactive components was based on some inclusion and exclusion criteria. The inclusion criteria were: (1) studies reported about the Ecuadorian medicinal plants and their medicinal uses for diabetes; (2) studies of medicinal plants with available scientific evidence of antidiabetics such as in vitro activities, in vivo studies, phytochemical analysis, clinical trials, and toxicological evidences published in scientific journals; and (3) studies reported about phytochemicals with in vitro or in vivo antidiabetic evidence. The studies with incomplete data, studies that did not elucidate antidiabetic properties or chemical composition, and studies with unavailability of in vitro or in vivo antidiabetic evidence were excluded from this review. Scientific names, families, and synonymous names of all selected plant species and all the taxonomic data used in this review were used according to the International Plant Names Index (https://www.ipni.org/), Plants of the World Online (https://powo.science.kew.org/) and the WFO Plant List (https://wfoplantlist.org/). All plants were sorted alphabetically according to botanical family and summarized in a final table.

## 3. Diabetic Burden and Its Impact on the Health and Economy of Ecuador

DM is the most chronic endocrine dysfunction and is often linked to hyperglycemic conditions and insulin deficiency, insulin resistance, or both [35,36]. Furthermore, it is currently among the top 10 major causes of mortality and disability worldwide, being dubbed as the epidemic of the 21st century [37]. Long-term DM may culminate in several complications like peripheral blood vessel disease, atherosclerosis, heart attacks, neuropathy, nephropathy, and retinopathy [2,38,39]. Approximately 422 million individuals in developing countries suffer from diabetes, with the majority residing in low- and middle-income nations, is solely accountable for more than 1.5 million deaths annually [40], and is expected to reach 700 million by 2045 [41]. It is also predicted that the prevalence rate of diabetes in Latin America and the Caribbean will rise to 11.3% [42]. According to IDF 2021 study, 4.7% of Ecuadorians have diabetes [7] and around 63% of the Ecuadorian population will be subjected to DM cases in the future [43]. Diabetes was Ecuador’s second most common cause of death in 2021 after COVID-19 [8]. Ecuador has a 40% poverty rate with fragmented health systems and high out-of-pocket health costs that impose financial burdens on individuals [7,9]. The cost of treating diabetes is 3747.44 USD per person, which represents 73% of the minimum annual income of an Ecuadorian employee [44]. Contemporary treatment for diabetes focuses on suppressing and controlling blood glucose levels through lifestyle changes, diet, and weight control. Antidiabetic medications, such as insulin injections and oral hypoglycemic drugs, are responsible for the side effects like hypoglycemia, lactic acid intoxication, and gastrointestinal upset. Efforts are being made to find suitable antidiabetic and antioxidant therapies to address these side effects. Also, antidiabetic drugs are scarce and expensive in Ecuador, which causes a huge financial burden during the post-treatment phase of diabetes. Due to the high prevalence and long-term complications of DM, such as obesity, edema, and cardiovascular threats [45], there is the utmost need for developing new oral hypoglycemic agents from herbal and medicinal plants. Previous studies by Chikhi et al. (2014) [46] confirmed high antidiabetic activities in *Atriplex halimus*. During the last few decades, medicinal plants like *Allium sativum*, *Eugenia jambolana*, *Momordica charantia*, *Ocimum sanctum*, *Phyllanthus amarus*, *Pterocarpus marsupium*, *Tinospora cordifolia*, *Trigonella foenum graecum*, and *Withania somnifera* that have proven antidiabetic properties were used for developing natural, safe, and cost-effective herbal antidiabetic drugs [47] and were supplemented with a plethora of bioactive compounds such as alkaloids, glycosides, terpenes, flavonoids, etc. [48]. However, Ecuadorian herbs like *Schinus molle*, *Opuntia soehrensii*, *Lepidium meyenii*, *Cyclanthera pedate*, *Smilax officinalis*, *Uncaria tomentosa*, etc., that are used in traditional medicine are reported to possess antidiabetic, antioxidant, anti-inflammatory, and anti-hypoglycemic activities [27]. Therefore, more such Ecuadorian plant species should be explored, focusing on their antidiabetic properties to develop cost-effective and safe antidiabetic herbal remedies which could be of great economic importance for the pharmaceutical industries in Ecuador.

## 4. Medicinal Plants with Antidiabetic Potential Used in the Traditional Medicine of Ecuador

Based on the literature survey, 27 plant species belonging to 23 botanical families were identified, of which 15 are native to Ecuador and 12 were introduced from other locations, which are used in the traditional medicine of Ecuador to treat hyperglycemia. The highest representation of species comes from families Asteraceae, Apiaceae, Costaceae, and Leguminosae, with two representatives each, followed by 19 families each containing one species. Figure 1 describes the botanical family distribution of 27 plant species. However, Asteraceae is the most encountered ethnobotanical family that represents the highest number of species in South America [49,50]. 

*Adiantum poiretii*, a perennial rhizomatous geophyte belonging to the Pteridaceae family, thrives mainly in the wet, tropical biome. Their native range extends to tropical and subtropical America, Nigeria, Ethiopia, South Africa, the Arabian Peninsula, the Indian Ocean, India, and Sri Lanka. It plays a vital role in the traditional medicine of Ecuador because of its medicinal and environmental benefits [23,51]. The methanol extract from *A. poiretii* was found to possess antidiabetic properties, revealing moderate inhibition activity against α-glucosidase, an enzyme responsible for reducing post-meal blood sugar spikes with calculated IC_50_ values of 46.3 μg/mL. However, no significant activity of α-amylase was observed in the study [26]. 

Another plant species, *Artocarpus altilis*, commonly referred to as breadfruit and classified under the Moraceae family, is recognized for its richness in carbohydrates and diverse beneficial properties including antibacterial, antitubercular, antiviral, antifungal, antiplatelet, anti-arthritis, and antidiabetic activities. Methanolic extract showed α-glucosidase inhibitory activity with an IC_50_ value of 40.9 μg/mL, whereas acarbose was used as a positive control with an IC_50_ value of 964.6 μg/mL [26,52].

*Baccharis genistelloides* is often identified as an herb or shrub within the Asteraceae family and is one of the native South American plants widely used in traditional medicine, often prepared through infusion, to which is attributed its anti-inflammatory properties [53]. A study by Jaramillo Fierro and Ojeda Riascos (2018) [26] revealed that *B. genistelloides* showed significant inhibitory activity against α-glucosidase enzymes, with an IC_50_ value of 154.6 μg/mL. This suggests the capability of *B. genistelloides* to modulate glucose metabolism and beckons further exploration within therapeutic contexts.

*Cajanus cajan*, also known as Fréjol de palo, is a perennial shrub in the Fabaceae family, native to Asia and widely used as a pulse in the region. Its extracts are globally employed for treating various ailments including diabetes, dysentery, hepatitis, and measles. Notably, the leaves of *Cajanus cajan* are abundant in flavonoids and stilbenes, which are responsible for their beneficial effects on human health [54]. The antidiabetic activity of *Cajanus cajan* was tested on alloxan-induced diabetic Swiss albino mice. The results revealed that orally administered doses of plant extract showed high potential for the reduction of blood glucose levels with a maximum reduction of 54.51% of blood glucose levels, and an IC_50_ value of 17.44 μg/mL was obtained [54].

*Costus comosus*, often known as red tower ginger, is a perennial plant originating from Ecuador, belonging to the Costaceae family. Beyond its aesthetic appeal, this rhizomatous plant has a longstanding history of medicinal applications. Traditionally, its leaves and rhizomes were utilized to address a variety of ailments such as fever, rash, asthma, bronchitis, intestinal worms, diabetes, and liver diseases. The study of antidiabetic properties of *C. comosus* was based on their inhibition abilities on α-glucosidase and α-amylase enzymes. A study by Jaramillo Fierro and Ojeda Riascos (2018) [26] revealed that the methanolic extract of *C. comosus* showed inhibitory activity only on α-glucosidase with an IC_50_ value of 57.9 μg/mL.

*Croton wagneri*, commonly known as moshquera blanca or moshquera, is a shrub belonging to the Euphorbiaceae family and endemic to the Andean region of Ecuador where it thrives abundantly. This shrub is documented in at least seven populations ranging from the Carchi province in the north to the Azuay province in the south [55]. The incorporation of this plant into food preparations is a promising approach for locals to enhance nutritional value. A study by Jaramillo Fierro and Ojeda Riascos (2018) [26] revealed that *C. wagneri* has a lower inhibitory effect on α-amylase (IC_50_ > 1000) and α-glucosidase with an IC_50_ value of 162.4 μg/mL.

*Foeniculum vulgare*, commonly known as fennel, stands as a widely recognized perennial herbaceous plant extensively employed in global herbal medicine and culinary practices. Belonging to the Apiaceae family, fennel has a rich history in ethnobotanical applications, addressing diverse health concerns such as gastrointestinal issues, hormonal disorders, and reproductive and respiratory diseases. Fennel provides flexible culinary choices in Ecuador, whether consumed raw, cooked, or baked [56]. The antihyperglycemic effect of the aqueous extract of *F. vulgare* was demonstrated in a study on streptozotocin-induced diabetic (STZ) rats resulting in reduced blood glucose levels after 6 h from administration and without any loss of body weight. Furthermore, it improved oral glucose tolerance in diabetic rats [57].

*Glycyrrhiza glabra*, a member of the Leguminosae family, holds a prominent position in the ancient medical traditions of Ayurveda, serving as both a medicinal herb and a flavoring agent. Commonly referred to as licorice, liquorice, or sweet wood, *G. glabra* is predominantly found in Mediterranean regions and specific areas of Asia [58]. This plant species has a traditional value in Ecuador where it is used for various purposes in food and traditional medicine preparation [59]. The ethanolic extract of *G. glabra* roots was introduced to streptozotocin-induced diabetic rats at doses of 200 mg/kg and 400 mg/kg, respectively. The results indicated a significant reduction of fasting blood glucose and fasting serum insulin in diabetic rats [60].

*Ilex guayusa*, commonly referred to as Guayusa, is an underexplored holly species belonging to the only extant genus of the family Aquifoliaceae and thriving in the upper Amazon basin across Colombia, Ecuador, and Peru. Guayusa holds cultural significance as societies in this region traditionally brew an infusion from its leaves for consumption [61]. A study by Jaramillo Fierro and Ojeda Riascos (2018) [26] revealed that *I. guayusa* showed moderate inhibitory activity with an IC_50_ value of 176.5 μg/mL for α-glucosidase inhibition, but its high antioxidant potential contributes to the treatment of diabetes due to its ability to counteract oxidative stress, a factor in the complications of diabetes.

*Ipomoea carnea*, commonly known as morning glory and a member of the Convolvulaceae family, is a shrub that holds a prominent place in Ecuadorian folk medicine for its historical use in managing diabetes and its muscle relaxant properties. Notably, it is well-known for its substantial antioxidant properties, boasting high levels of phenolic compounds, flavonoids, and tannins, which underscores its potential therapeutic values [62]. Research on *I. carnea* leaf extracts explored their phenolic, flavonoid, and tannin contents alongside their potentials for treating diabetes in Wistar rats. After three weeks of oral administration, both the aqueous and alcoholic extracts precipited a decrease in blood glucose levels, with the alcoholic extract demonstrating superior efficacy. Furthermore, a reduction in body weight was also observed [62].

*Justicia colorata*, locally known as Insulina, is a shrub indigenous to Ecuador and Peru, belonging to the Acanthaceae family. This plant is commonly consumed in Ecuador through the infusion method. The antidiabetic effect of *J. colorata* was confirmed in a study in Ecuador focusing on in vitro hypoglycemic and antioxidant activities. The results showed α-glucosidase inhibition activity with an IC_50_ value of 622.1 μg/mL [26].

Chamomile (*Matricaria chamomilla*), a prized medicinal herb native to Europe and belonging to the Asteraceae family, is renowned as a “star among medicinal species”. Chamomile is widely known for its broad popularity and extensive use in the folk and traditional medicine of Ecuador [63,64]. The antidiabetic activity of *M. chamomilla*’s ethanolic extract of the aerial part was studied in streptozotocin-induced diabetic rats. This study revealed that the administration of varying doses of *M. chamomilla* led to a notable decrease in postprandial hyperglycemia, oxidative stress, and the enhancement of the antioxidant system [65].

*Neonelsonia acuminata*, native to Mexico, Venezuela, and Peru, is a perennial herb within the Apiaceae family. These plants grow primarily in the subalpine or subarctic biome. Their roots are employed in the traditional medicine of Ecuador and are often consumed in their raw form [23]. The inhibitory effects of *N. acuminata* on α-glucosidase and α-amylase were analyzed in this study. The results showed activity only on α-glucosidase with IC_50_ values of 198.7 μg/mL, introducing an inhibitory capacity similar to established drugs used in diabetes treatment. Additionally, *N. acuminata* displayed remarkable free radical scavenging activity (DPPH), indicating its prospective role as an alternative enzyme inhibitor and antioxidant in the management of DM [26].

*Oreocallis grandiflora*, a species native to Ecuador within the Proteaceae family, is commonly known as cucharillo, cucharilla, gañal, and algil. Traditionally, their leaves and flowers were harvested during the blooming phase and were orally administered for the treatment of liver disease, vaginal bleeding, and inflammation of the ovaries and uterus. Additionally, it is employed as a remedy for digestive issues, diuretic effects, and hypoglycemic conditions [66]. A study performed by Jaramillo Fierro and Ojeda Riascos (2018) [26] revealed its impressive ability to slow down both α-amylase and α-glucosidase enzymes, with an IC_50_ value of 161.5 μg/mL for α-amylase and an incredibly low IC_50_ value of 2.8 μg/mL for α-glucosidase inhibition. These values signify its strong capability for inhibiting these enzymes, especially α-glucosidase, which is crucial for regulating blood sugar after meals. Moreover, its notable antioxidant activity suggests that it could be valuable in combating oxidative stress, a significant factor in diabetes-related complications.

*Pelargonium graveolens*, a kind of herb within the Geraniaceae family, is recognized for its aromatic properties. Its global cultivation primarily focuses on the extraction of essential oils employed mainly in the perfume industry due to its highly desirable scent [67]. In Ecuador, this plant species is commonly used in traditional medicine. The essential oil from *P. graveolens* was evaluated using an α-glucosidase inhibition assay. The findings showed a promising efficiency of *P. graveolens* essential oil in inhibiting α-glucosidase enzyme with an IC_50_ value of 93.72 μg/mL similar to the IC_50_ value of acarbose, which was 80.4 μg/mL. Therefore, its efficiency in reducing post-meal blood sugar spikes can be an effective remedy for diabetes [68].

Avocado (*Persea americana*), native to Mexico and belonging to the Lauraceae family, has surged in popularity as a “superfood” for its unique nutritional profile and health benefits. In addition to its culinary uses, it is traditionally used in Ecuador for medicinal purposes including blood pressure and blood sugar management, antiviral properties, and cardiovascular health. A study by Abd Elkader et al. (2022) [69] demonstrated that the ethanolic extracts of avocado fruit exhibited higher α-amylase inhibition activity. Another study by Ojo et al. (2022) [24] exhibited maximal α-glucosidase inhibitory activity of 56.41% in alloxan-induced diabetic male Wistar rats injected with aqueous extract of *P. americana* seeds and 21.42% of α-amylase inhibitory activity. Another study showed that the aqueous extracts of *P. americana* seed were administered to DM-induced male Wistar rats at doses of 26.7, 53.3, and 106.6 mg/kg body weight. The results indicated a marked decrease in fasting blood glucose levels as well as TG, LDL-c, G6P, F-1, 6-BP, MDA, IL-6, TNF-α, and NF-ĸB as well as an increase in liver glycogen, hexokinase, and HDL-c [70].

*Physalis peruviana*, also called golden berry or Cape gooseberry, is a tropical herb from the Solanaceae family. The plant extract of this species has a history of use in folk medicine for treating diabetes, hepatitis, malaria, dermatitis, asthma, and rheumatism [71]. The aqueous extract of the dried leaf powder of *P. peruviana* led to a notable decrease in blood sugar levels in guinea pigs at a dose of 100 mg/kg. However, higher doses caused intoxication [72].

*Piper crassinervium*, known as Guabiduca in Ecuador, is a shrub in the Piperaceae family found throughout South America. It has a rich value in traditional medicine in Ecuador for its valuable essential oil that makes it commercially viable [73]. A study by Jaramillo Fierro and Ojeda Riascos (2018) [26] confirmed that the extract from *P. crassinervium* effectively brings down glucose absorption and suppresses postprandial hyperglycemia. The results revealed the ability to inhibit the α-glucosidase enzyme with an IC_50_ value of 108.5 μg/mL, which helps to break down complex sugars into simple forms.

*Ruta graveolens*, commonly known as rue, is a subshrub from the Rutaceae family that originated in the Mediterranean but is now grown worldwide. It has been integral to European pharmacopeia since ancient times. Extracts and essential oils from rue are utilized in drug development due to their diverse pharmacological benefits, including antibacterial, analgesic, anti-inflammatory, antidiabetic, and insecticidal properties. Traditionally, it has been used in Ecuador for pain relief, eye problems, rheumatism, and dermatitis [74]. The antidiabetic properties of *R. graveolens* were studied using α-glucosidase and α-amylase inhibition assay. Findings revealed that methanolic extract showed higher inhibitory activity of α-glucosidase and α-amylase with an IC_50_ value of 281 and 215 μg/mL, respectively, which showed better activity than the reference drug acarbose [75].

*Siparuna eggersii*, a shrub belonging to the Monimiaceae family, is endemic to the Loja province in Ecuador and is locally known as “Monte del oso”. It has a high ethnic value in equatorial regions, but this species is currently facing the threat of extinction [76]. The inhibitory activity of *S. eggersii* on α-glucosidase and α-amylase was investigated in a study that demonstrated notable inhibitory activity against α-glucosidase enzymes, revealing an IC_50_ value of 28.3 μg/mL. This finding suggests the compelling potential of *S. eggersii* for diabetes treatment [26].

*Urtica dioica* L., commonly known as nettle, is a perennial flowering plant originating in Europe, temperate Asia, and western North Africa, with widespread cultivation across the globe [77]. As a member of the Urticaceae family, nettle is recognized for its valuable bioactive compounds, including formic acid and abundant flavonoids, making it an important subject in phytotherapy. Numerous studies have highlighted its therapeutic potential for treating prostatic hyperplasia, rheumatoid arthritis, allergies, anemia, internal bleeding, kidney stones, burns, and infectious diseases, supported by studies showcasing its anti-proliferative and antimicrobial activities [78]. The antihyperglycemic activity of the aqueous extract of *U. dioica* was demonstrated in a study by Bnouham et al. (2003) [79] on male Wistar rats and Swiss mice of both sexes. The results indicated a decrease in glycemia by approximately 33% compared to the control, during the first hour after glucose insertion. Furthermore, the reduction of intestinal glucose absorption was observed in this study.

*Verbena litoralis*, a South American herb in the Verbenaceae family, is historically used in traditional medicine for diarrhea, fever, gastrointestinal issues, and sexually transmitted diseases. This plant attracted more attention for its nerve-growth-factor potentiating activity [80]. A study by Jaramillo Fierro and Ojeda Riascos (2018) [26] elucidated the inhibitory activity of the key enzyme α-glucosidase that is responsible for breaking down sugars. The results exhibited that *V. litoralis* extract showed inhibitory activity on α-glucosidase with IC_50_ values of 337.9 μg/mL. However, the extract was found ineffective for α-amylase inhibition.

Out of 27 Ecuadorian plant species, 22 of them have rational reports supporting antidiabetic properties that were pharmacologically elucidated. However, no existing pharmacological studies have provided evidence supporting the antidiabetic activities of *Bryophyllum gastonis*-*bonnieri*, *Costus villosissimus*, *Juglans neotropica*, *Pithecellobium excelsum*, and *Myroxylon peruiferum*. However, these plants are used in Ecuadorian traditional medicine to treat diabetes [34,59,81,82]. Ecuadorian medicinal plants are rich in valuable bioactive phytochemicals such as flavonoids, phenolics, fatty acids, aldehydes, and other components that are involved in the pharmacological mechanism of antidiabetic activity, act as regulators for insulin secretion, sensitivity, and insulin-signaling pathways and enhance glucose uptake as well as modulate intestinal glucose absorption and reduce oxidative stress and inflammation [29,83,84,85]. Several plants used in Ecuadorian traditional medicine for treating diabetes are also widely used in other systems of medicine. Among 27 elucidated Ecuadorian plants, 15 are native to Ecuador and the remaining 12 were introduced. *F. vulgare* is a well-known antidiabetic plant used in traditional Egyptian medicine [86] and Ayurveda [87]. The use of *M. chamomilla* is also well-established in Indian, European, and Egyptian medicine [88]. *C. cajan* is traditionally used as medicine in India, China, America, and Europe [89]. *P. americana* is widely used in European and American countries for its antidiabetic properties [90]. Similarly, the use of *G. glabra* extract is extensively mentioned in traditional Chinese medicine and Ayurveda [25]. *A. poiretii* is frequently used in Ayurveda and Unani [91]. In American and African traditional medicine, *P. peruviana* is extensively used [92]. Similarly, the use of *U. dioica* leaves in the traditional practice of Ayurveda and Arabic traditional medicine is well-documented [93]. Additionally, herbal recipe analysis revealed that, unlike other medicinal systems, polyherbal mixtures of different plants are mainly used for traditional medicine preparation as remedies for diabetes. Infusion is the most common preparation method used, followed by decoction, juice, crushed fresh, cooked, and eaten raw. The plant part used for this process was most frequently leaf and stem, closely followed by bark, whole plant, aerial part, flower, fruit, root, and seed (Table 1) [34]. Table 1 summarizes the list of medicinal plants with antidiabetic potentials used in the traditional medicine of Ecuador. A map representing the distribution of these 27 medicinal plants with antidiabetic properties is shown in Figure 2.

## 5. Bioactive Compounds of Ecuadorian Medicinal Plants and Their Anti-Diabetic Properties

Due to the systemic toxic exposure and limited efficacy of current antidiabetic medication, new plant-based natural antidiabetic drugs are in utmost need to be discovered. To circumvent these hurdles, a plant-based medication approach can be adopted as a sustainable alternative antidiabetic agent. The phytochemicals are mainly responsible for the biological activity of the plant extracts [98]. Phytochemicals found in plant extracts are the major contributors with various anti-inflammatory, antioxidant, and antidiabetic properties, which are effective, economical, and low in toxicity [99]. The activity of phytochemicals-based drugs not only increases the antidiabetic activity but also acts as an enhanced anti-inflammatory and antioxidant agent that can dramatically improve the targeting of the interaction site while minimizing systemic exposure and associated toxicity. Medicinal plants are key sources of producing a variety of phytochemicals such as alkaloids, phenolic acids, flavonoids, glycosides, saponins, polysaccharides, stilbenes, and tannin, which have immense antidiabetic effects that are employed through various mechanisms such as the regulation of glucose and lipid metabolisms, insulin secretion, stimulation of β cells, NF-kB signaling pathway, inhibition of gluconeogenic enzymes, and reactive oxygen species (ROS) protective action [29].

Ecuadorian medicinal plants have a plethora of phytochemicals that have in vitro and in vivo evidence in support of antidiabetic properties. Flavonoids are the major bioactive compounds found in most of the Ecuadorian medicinal plants that possess antidiabetic properties. Due to their hydroxyl groups and aromatic rings, they serve as natural antioxidants [29]. Flavonoids such as cirsimaritin, cirsiliol, hispidulin, genkwanin, and apigenin are the major bioactive compounds found in *Baccharis genistelloides* [53]. A study by Alqudah et al. (2023) [100] showed the importance of cirsimaritin in type-2 diabetes. They observed reduced levels of serum glucose and a reduction in the increased serum insulin level in diabetic rats. Another study by Escandón-Rivera et al. (2019) [101] showed hypoglycemic effect of *Bromelia karatas* on STZ-NA-induced diabetic rats, and the chemical composition analysis revealed cirsiliol as one of the major compounds responsible for this activity. Another flavone hispidulin confirmed the stimulation of glucagon-like peptide-1 and suppressed hepatic glucose production in STZ-NA-induced diabetic rats [102]. Genkwanin is also a potent antioxidant that is responsible for α-amylase inhibitory activity [103]. A study by Ihim et al. (2023) [104] reported that apigenin facilitates glucose-stimulated insulin secretion and prevents endoplasmic reticulum (ER) stress-mediated β-cell apoptosis in the pancreas. Other flavones such as pinostrobin, genistein, quercetin, quercetin-3-O-hexose, myricetin 3-O-β-glucuronide, fisetin, isorhamnetin hexuronide, quercetin 3-O-rutinoside, quercetin 3-O-β-glucuronide, isorhamnetin hexoside, and isorhamnetin 3-O-rutinoside are the major bioactive compounds that have antidiabetic properties found in *Cajanus cajan* [89], *Ilex guayusa* [105], *Oreocallis grandiflora* [66], *Pelargonium graveolens* [106], *Verbena litoralis* [107], and *Bryophyllum gastonis-bonnieri* [108].

Similarly, plant-derived phenols are bioactive compounds that possess immense antidiabetic activities through various mechanisms such as AMPK pathway activation, α glucosidase or α amylase inhibition, glucose regulation, insulin sensitivity improvement, and PPAR activation [83]. Chlorogenic acid, caffeic acid, coumaric acid, ellagic acid, ferulic acid, and vanillic acid are the natural phenols found in many Ecuadorian plants such as *Ilex guayusa* [105], *Pelargonium graveolens* [106], *Persea americana* [31,109], *Verbena litoralis* [107], *Artocarpus altilis* [110], and *Bryophyllum gastonis-bonnieri* [108]. The antidiabetic activity of chlorogenic acid is attributed to an increase in the glucose uptake in L6 muscular cells and a rise in insulin secretion from the INS-1E insulin-secreting cell line and rat islets of Langerhans [111]. Another natural phenol coumaric acid showed antidiabetic activities by lowering the blood glucose level and gluconeogenic enzymes and increasing the activities of hexokinase, glucose-6 phosphatase dehydrogenase, and GSH via increasing the level of insulin [97]. A study by Narasimhan et al. (2015) [112] demonstrated that ferulic acid improves insulin sensitivity and hepatic glycogenesis, inhibits gluconeogenesis, and maintains insulin signaling to maintain normal glucose homeostasis. Similarly, caffeic acid was reported to reduce hepatic glucose output and to enhance adipocyte glucose uptake, insulin secretion, and antioxidant capacity [113]. Singh et al. (2022) [114] found reduced hyperglycemia and GSH and increased liver enzymes in diabetic rats as protective effects of vanillic acid.

Another class of the important bioactive group terpenoids are the major compounds found in many natural products, and several terpenoids have been reported as antidiabetic agents. Some of them are under various stages of pre-clinical and clinical evaluation to develop them as antidiabetic agents. These compounds mainly act as enzyme inhibitors that are responsible for the development of insulin resistance and the normalization of plasma glucose and insulin levels, and play an important role in glucose metabolism by inhibiting several pathways involved in diabetes and associated complications [101]. Numerous research studies have reported that bioactive terpenoids are associated with anti-diabetic properties. Betulinic acid, camphene, myrcene, fenchone, limonene, α-phellandrene, carvone, spathulenol, caryophyllene oxide, germacrene D, α-pinene, β-caryophyllene, α-humulene, catechin, phytol, carvacrol, hexahydrofarnesyl acetone, and (E)-β-ionone are found in various Ecuadorian medicinal plants; for instance, *Cajanus cajan* [89], *Costus comosus* [114], *Croton wagneri* [55], *Foeniculum vulgare* [115], *Glycyrrhiza glabra* [116], *Ipomoea carnea* [30], *Myroxylon peruiferum* [117], *Persea americana* [31,109], *Physalis peruviana* [118], *Matricaria chamomilla* [119], *Piper crassinervium* [120], *Siparuna eggersii* [74], and *Urtica dioica* [120], which are traditionally used for their antidiabetic properties.

Fatty acids represent significant bioactive compounds capable of serving as potent antidiabetic agents by reducing blood cholesterol levels, stabilizing lipid and protein metabolisms, enhancing the liver’s detoxification function, and stimulating immune-protective mechanisms. Additionally, they increase the elasticity of blood vessel walls, decrease permeability, enhance microcirculation, and inhibit the oxidation of cell membrane lipids [85]. n-Hexadecanoic acid is a plant-based fatty acid mainly found in several Ecuadorian plants such as *Artocarpus altilis* [110], *Costus comosus* [121], and *Glycyrrhiza glabra* [116]. It is a potent antioxidant agent and can inhibit α-amylase and α-glucosidase activity that are responsible for glucose production in the body [122]. Another natural fatty acid dodecanoic acid also acts as a natural antidiabetic agent found in *Glycyrrhiza glabra* [116]. It was found that dodecanoic acid (lauric acid) decreases fasting blood glucose levels and induces β-cell regeneration in high-fat diet/streptozotocin-induced type 2 diabetic male Wistar rats [123].

2-Heptadecenal is an aldehyde found in *Artocarpus altilis* [95], which acts as a natural inhibitor of the α-amylase and α-glucosidase enzymes. A study by Natta et al. (2023) [124] reported antiglycemic properties of *Artocarpus heterophyllus*, in which they found 2-heptadecenal to be the major compound responsible for this activity. Anethole is another phenylpropanoid found in *Urtica dioica* [125] and *Foeniculum vulgare* [115]. This bioactive compound is reported to suppress diabetic nephropathy in streptozotocin-induced diabetic rats by decreasing blood glucose levels and downregulating the angiotensin II receptor (AT1R) and TGF-β1 expression [126]. The phenylpropene, namely, estragole (methyl chavicol), acts as an enzyme inhibitor against α-amylase, lipase, and tyrosinase and serves as a potent antioxidant agent that can improve diabetes [127,128]. This bioactive compound is also found in *Foeniculum vulgare* [103]. n-Hexacosane is a natural alkane that improves blood glucose levels, glucose tolerance, glycated hemoglobin, and liver glycogen [129] found in *Physalis peruviana* [118]. The polysaccharide holocellulose and organic polymer lignin are found in *Juglans neotropica* [130]. Holocellulose mainly acts as an activator for glucokinase in reducing blood sugar [131], and lignin acts as a natural antioxidant and can increase the inhibitory effect on α-amylase [132]. Another natural polycyclic aromatic hydrocarbon naphthalene found in *Urtica dioica* [125] can act as a potential anti-hyperglycemic agent by inhibiting the α-glucosidase enzyme [133]. In conclusion, we found that Ecuadorian medicinal plants are rich in phytochemicals that can be a potential antidiabetic agent. Table 2 summarizes the list of active compounds with antidiabetic properties found as major compounds in the Ecuadorian medicinal plants that are mainly used as traditional medicines against diabetes.

## 6. Conclusions and Future Perspectives

DM is one of the major health burdens of Ecuador compounded by high out-of-pocket health costs subjecting Ecuadorians with diabetes who want effective treatment to substantial financial strain. In this review, we explore and elucidate the Ecuadorian medicinal plants that are traditionally used for treating diabetes and are natural, safe, non-GMO, cost-effective, and low in toxicity. We elucidated 27 of the above-mentioned plants, which are a fundamental part of the health systems of several Ecuadorian ethnic groups. Immense anti-diabetic properties were found in Ecuadorian plant species such as *I. guayusa*, *A. poiretii*, *A. altilis*, *J. colorata*, *N. acuminata*, *V. litoralis*, *R. graveolens*, *O. grandiflora*, *P. crassinervium*, *S. eggerssii*, *P. graveolens*, *C. wagneri*, and *C. comosus*; these have α-glucosidase and/or α-amylase inhibitory activities, which are the key enzymes for carbohydrate metabolism. Similarly, *A. altilis*, *B. genistelloides*, *F. vulgare*, *I. carnea*, *M. chamomilla*, *C. cajan*, *P. americana*, *G. glabra*, and *U. dioica* were found to exhibit high antidiabetic potential during the clinical trials conducted in animals. The contemporaneous occurrence of an extensive range of phytochemicals such as alkaloids, phenolic acids, flavonoids, glycosides, saponins, polysaccharides, stilbenes, tannin, etc., are the main bioactive compounds present in these Ecuadorian medicinal plants that are responsible for high antidiabetic properties and are often hallmarks of the practice of phytotherapy. However, the bioactive compounds directly associated with the antidiabetic activities of some plants are still unexplored. Apart from these, plants like *B. gastonis-bonnieri*, *C. villosissimus*, *J. neotropica*, *P. excelsum*, and *M. peruiferum* that are used as a part of folklore medication for treating diabetes, are deprived of pharmacological studies to elucidate their antidiabetic activity.

Comprehending the application of Ecuadorian plants in the treatment of diabetes poses numerous obstacles. Several Ecuadorian plant species used as traditional medication against diabetes lack rational elucidation of their phytochemical constituents, cytotoxicity analyses, pharmacological activities, and clinical trials. In the cases of *J. colorata* and *A. poiretii*, in which the preclinical experimental reports supported antidiabetic activity, the identification and purification of bioactive components, pharmacodynamics, pharmacokinetics, bioavailability, and bioactivity should be conducted to understand the metabolism of the bioactive compounds. Standardized formulation should be introduced to reduce the production-batch heterogeneity caused by geographical and environmental factors. Because the pharmacological studies on *P. graveolens*, *C. comosus*, and *O. grandiflora* showed an optimistic outlook, more robust clinical studies using rats and humans, and the investigation of standardized extracts, are required to analyze the efficacy from both traditional use and potential use perspectives. Also, the determination of the long-term side effects of natural herbal product formulations individually and in combination with synthetic drugs is required. Furthermore, large-population human clinical evaluations are needed before diabetic patients can rely solely on plant-based therapies for controlling diabetes with no side effects. 

The 27 Ecuadorian medicinal plants elucidated in this review hold promise as potential antidiabetic medications. that can add great economic value to the pharmaceutical and food industries in Ecuador and other developing countries. However, further studies would confirm their practical application in the above-mentioned industries.

## Figures and Tables

**Figure 1 plants-13-01436-f001:**
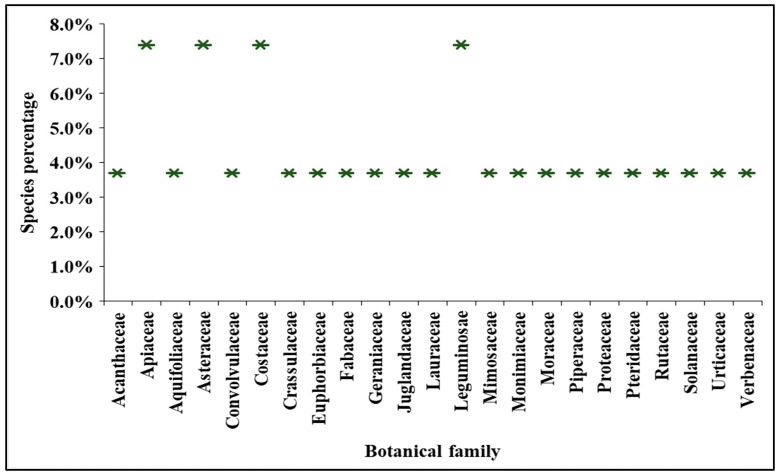
Family representation of 27 Ecuadorian medicinal plants used for the treatment of diabetes.

**Figure 2 plants-13-01436-f002:**
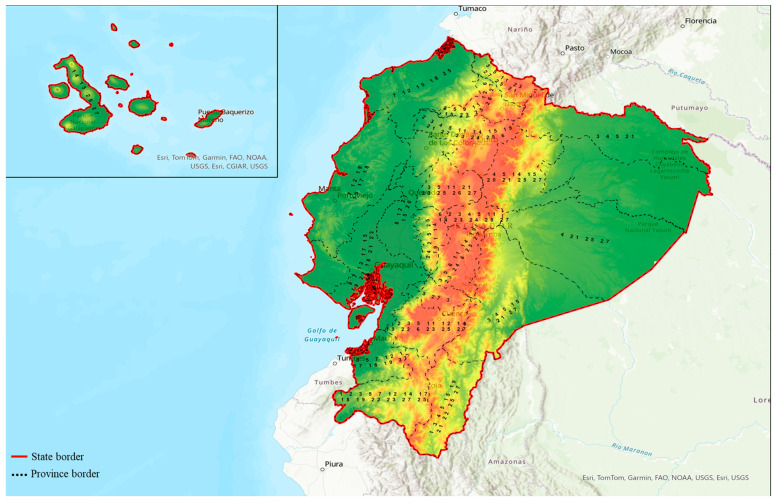
A simplified map showing the distribution of 27 medicinal plants with antidiabetic properties in Ecuador. 1-*Justicia colorata* (Nees) Wassh, 2-*Foeniculum vulgare* Mill., 3-*Neonelsonia acuminata* (Benth.) J.M.Coult. and Rose ex Drude, 4-*Ilex guayusa* Loes., 5-*Baccharis genistelloides* (Lam.) Pers., 6-*Matricaria chamomilla* L., 7-*Ipomoea carnea* Jacq., 8-*Costus comosus* Roscoe, 9-*Costus villosissimus* Jacq., 10-*Bryophyllum gastonis-bonnieri* (Raym.-Hamet and H.Perrier) Lauz.-March., 11-*Croton wagneri* Müll.Arg., 12-*Cajanus cajan* (L.) Millsp., 13-*Pelargonium graveolens* L’Hér., 14-*Juglans neotropica* Diels, 15-*Persea americana* Mill., 16-*Glycyrrhiza glabra* L., 17-*Myroxylon peruiferum* L.f., 18-*Pithecellobium excelsum* (Kunth) Mart., 19-*Siparuna eggersii* Hieron., 20-*Artocarpus altilis* (Parkinson) Fosberg, 21-*Piper crassinervium* Kunth, 22-*Oreocallis grandiflora* R.Br., 23-*Adiantum poiretii* Wikstr., 24-*Ruta graveolens* L., 25-*Physalis peruviana* L., 26-*Urtica dioica* L., 27-*Verbena litoralis* Kunth.

**Table 1 plants-13-01436-t001:** Ecuadorian medicinal plants used in traditional medicine to treat diabetes.

Family	Scientific Name	Local Name (Vernacular Name)	Plant Part Used	Traditional Preparations	Antidiabetic Activity		References
					In Vitro	In Vivo	
Acanthaceae	*Justicia colorata* (Nees) Wassh.	Insulina	Leaf, stem	Infusion	Methanolic extract showed α-glucosidase inhibitory activity with an IC_50_ value of 622.1 μg/mL where acarbose was used as a positive control, with an IC_50_ value of 964.6 μg/mL.	-	[26]
Apiaceae	*Foeniculum vulgare* Mill.	Hinojo, *eneldo*	Whole plant	Infusion	Aqueous leaf extract *F. vulgare* showed free-radicals scavenging activity with an IC_50_ value of 43 μg/mL, whereas synthetic antioxidant BHT showed an IC_50_ value of 22.67 μg/mL.	Aqueous leaf extract *F. vulgare* at a dose of 10 mg/kg reduced the blood glucose level in both normal and streptozotocin-induced diabetic rats. Also, it improved oral glucose tolerance in diabetic rats and revealed a positive effect on liver histology	[56,57,94]
	*Neonelsonia acuminata* (Benth.) J.M. Coult. and Rose ex Drude	Zanahoria blanca	Root	Eaten raw	Methanolic extract showed α-glucosidase inhibitory activity with an IC_50_ value of 198.7 μg/mL where acarbose was used as a positive control, with an IC_50_ value of 964.6 μg/mL.	-	[23,26]
Aquifoliaceae	*Ilex guayusa* Loes.	Guayusa	Leaf	Infusion	Methanolic extract showed α-glucosidase inhibitory activity with an IC_50_ value of 176.5 μg/mL where acarbose was used as a positive control, with an IC_50_ value of 964.6 μg/mL.	-	[26,61]
Asteraceae	*Baccharis genistelloides* (Lam.) Pers.	Tres filos	Aerial part	Aqueous infusion	Methanolic extract showed α-glucosidase inhibitory activity with an IC_50_ value of 154.6 μg/mL where acarbose was used as a positive control, with an IC_50_ value of 964.6 μg/mL.	-	[26,51,95]
	*Matricaria chamomilla* L.	Chamomile	Whole plant	Infusion	-	Ethanolic extract of *M. chamomilla* was introduced in streptozotocin-induced diabetic rats at doses of 5, 20, 50, and 100 mg/kg. Treatment with varying doses of MCE notably mitigated postprandial hyperglycemia and oxidative stress, bolstering the antioxidant system, while safeguarding pancreatic islet cells in histological examinations as compared to the control group.	[63,64]
Convolvulaceae	*Ipomoea carnea* Jacq.	Borrachera, matacabra	Aerial part	Infusion	-	Hot, cold, aqueous, and alcoholic extracts of *I. carnea* leaves were introduced to streptozotocin-induced diabetic Wistar rats at a dose rate of 500 mg/kg body weight. The results indicated that the alcoholic extract had more potential to lower blood glucose and lipid levels in diabetic rats.	[34,62]
Costaceae	*Costus comosus* Roscoe	Caña	Stem	Decoction	Methanolic extract showed α-glucosidase inhibitory activity with an IC_50_ value of 57.9 μg/mL whereas acarbose was used as positive control, with an IC_50_ value of 964.6 μg/mL.	-	[26,51]
	*Costus villosissimus* Jacq.	Caña agria	Leaf, stem	Infusion	Not elucidated	Not elucidated	[82]
Crassulaceae	*Bryophyllum gastonis-bonnieri* (Raym.-Hamet and H. Perrier) Lauz.-March.	Dulcamara	Leaf	Juice, crushed	Not elucidated	Not elucidated	[59,81]
Euphorbiaceae	*Croton wagneri* Müll.Arg.	Moshquera	Leaf	Aqueous infusion	Methanolic extract showed α-glucosidase and α-amylase inhibitory activities with IC_50_ values of 162.4 μg/mL and more than 1000 μg/mL, respectively, where acarbose was used as a positive control, with IC_50_ values of 964.6 μg/mL and 56.8 μg/mL, respectively.	-	[26,34,62]
Fabaceae	Cajanus cajan (L.) Millsp.	Fréjol de palo	Bark	Infusion	-	The antidiabetic activity of *C. cajan* methanolic extract was evaluated in alloxan-induced diabetic Swiss albino mice at doses of 200 and 400 mg/kg body weight, respectively. The results indicate a significant decrease in fasting serum glucose level and a reduction in blood glucose level during a 5 day study, as observed in the alloxan-induced diabetic mice.	[34,54]
Geraniaceae	*Pelargonium graveolens* L’Hér.	Esencia de rosa	Flower, leaf, stem	Infusion	The essential oil of *P. graveolens* leaves showed α-glucosidase inhibitory activity with an IC_50_ value of 93.72 μg/mL, whereas acarbose was used as the positive control with an IC_50_ value of 80.4 μg/mL.	-	[23,68]
Juglandaceae	*Juglans neotropica* Diels	Nogal, tocte	Leaf	Infusion	Not elucidated	Not elucidated	[34]
Lauraceae	*Persea americana* Mill.	Aguacate	Leaf, fruit, seed	Aqueous infusion, decoction	Aqueous extract of *P. americana* seed exhibited α-glucosidase and α-amylase inhibitory activities at 56.41% and 21.42%, respectively, whereas the positive control acarbose exhibited inhibitory activities of 76.41% for both enzymes.	Aqueous extracts of *P. americana* seed were administered to DM-induced male Wistar rats at a dose of 26.7, 53.3, and 106.6 mg/kg body weights, respectively. The results indicated a decrease in fasting blood glucose levels as well as TG, LDL-c, G6P, F-1, 6-BP, MDA, IL-6, TNF-α, and NF-ĸB. An increase in liver glycogen, hexokinase, and HDL-c was indicated.	[24,51,70]
Leguminosae	*Glycyrrhiza glabra* L.	Zaragoza	Leaf, stem	Infusion	-	Ethanolic extract of *G. glabra* roots was introduced to streptozotocin-induced diabetic rats at doses of 200 mg/kg and 400 mg/kg, respectively. The results indicate a significant reduction of fasting blood glucose and fasting serum insulin in diabetic rats.	[25,59,60]
	*Myroxylon peruiferum* L.f.	Bálsamo, chaquino	Bark	Infusion	Not elucidated	Not elucidated	[34]
Mimosaceae	*Pithecellobium excelsum* (Kunth) Mart.	Chaquiro	Bark	Infusion	Not elucidated	Not elucidated	[34]
Monimiaceae	*Siparuna eggersii* Hieron.	Monte del oso	Leaf	Crushed, infusion	Methanolic extract showed α-glucosidase inhibitory activity with an IC_50_ value of 28.3 μg/mL where acarbose was used as a positive control, with an IC_50_ value of 964.6 μg/mL.	-	[26,51]
Moraceae	*Artocarpus altilis* (Parkinson) Fosberg	Fruto del pan	Leaf	Aqueous infusion	Methanolic extract showed α-glucosidase inhibitory activity with an IC_50_ value of 40.9 μg/mL where acarbose was used as a positive control, with an IC_50_ value of 964.6 μg/mL.	-	[26,51,52]
Piperaceae	*Piper crassinervium* Kunth	Guabiduca	Stem, leaf	Decoction	Methanolic extract showed α-glucosidase inhibitory activity with an IC_50_ value of 108.5 μg/mL where acarbose was used as a positive control, with an IC_50_ value of 964.6 μg/mL.	-	[26,51]
Proteaceae	*Oreocallis grandiflora* R.Br.	Cucharillo	Leaf, bark, flower	Aqueous infusion	Methanolic extract showed α-glucosidase and α-amylase inhibitory activities with IC_50_ values of 2.8 μg/mL and 161.5 μg/mL, respectively, where acarbose was used as a positive control with IC_50_ values of 964.6 μg/mL and 56.8 μg/mL, respectively.	-	[23,26,51]
Pteridaceae	*Adiantum poiretii* Wikstr.	Culantrillo	Aerial part	Aqueous infusion	Methanolic extract showed α-glucosidase inhibitory activity with an IC_50_ value of 46.3 μg/mL where acarbose was used as a positive control with an IC_50_ value of 964.6 μg/mL.	-	[23,26,51]
Rutaceae	*Ruta graveolens* L.	Ruda	Stem, leaf	Infusion	Methanolic and chloroform extracts of *R. graveolens* showed α-glucosidase and α-amylase inhibitory activities with IC_50_ values of 281 and 460.5 μg/mL, and 215 and 479 μg/mL, respectively, whereas acarbose showed inhibitory activities of 484.2 and 69.7 μg/mL, respectively.	-	[75]
Solanaceae	*Physalis peruviana* L.	Uvilla, uchuva, uvilla lanuda	Fruit	Juice	-	Aqueous decoctions of *P. peruviana* leaf powder were administrated to guinea pigs at the dose range of 100 mg/kg to 3.2 g/kg of body weight. The dose of 100 mg/kg of aqueous extract induced a significant reduction of glucose but at doses exceeding 400 mg, alterations in blood, kidney, and liver markers were noted, with mortality observed at doses above 800 mg/kg and LD_50_ of approximately 1280 mg/kg was obtained.	[23,72,73]
Urticaceae	*Urtica dioica* L.	Ortiga	Whole plant	Infusion, fresh	-	Aqueous extract of *U. dioica* (250 mg/kg) showed a strong glucose-lowering effect in alloxan-induced diabetic rats. The decrease of glycemia has reached 33% of the control value 1 h after glucose loading.	[79,96]
Verbenaceae	*Verbena litoralis Kunth*	Verbena	Whole plant	Cooked, infusion	Methanolic extract showed α-glucosidase inhibitory activity with an IC_50_ value of 337.9 μg/mL where acarbose was used as a positive control, with an IC_50_ value of 964.6 μg/mL.	-	[26,97]

**Table 2 plants-13-01436-t002:** Bioactive phytochemicals of Ecuadorian medicinal plants and their antidiabetic properties.

Scientific Name	Bioactive Compounds with Antidiabetic Properties	Bioactive Chemical Group	Antidiabetic Properties	Reference
*Adiantum poiretii*	No proper evidence found	-	-	-
*Artocarpus altilis*	n-Hexadecanoic acid	Fatty acid	α-amylase and α-glucosidase inhibitory activity and antioxidant activity	[122]
	Ellagic acid	Phenol	α-amylase inhibitory activity and antioxidant activity, stimulate insulin secretion and decrease glucose intolerance	[134]
	2-Heptadecenal	Aldehyde	α-amylase and α-glucosidase inhibitory activity	[124]
*Baccharis genistelloides*	Cirsimaritin	Flavonoid	Reduces elevated levels of serum glucose in diabetic rats and abrogates the increase in serum insulin	[100]
	Cirsiliol	Flavonoid	Hypoglycaemic effect	[101]
	Hispidulin	Flavonoid	Stimulates glucagon-like peptide-1 and suppresses hepatic glucose production	[102]
	Genkwanin	Flavonoid	α-amylase inhibitory activity and antioxidant activity	[103]
	Apigenin	Flavonoid	Facilitates glucose-stimulated insulin secretion and prevents ER stress-mediated β-cell apoptosis in the pancreas	[104]
*Bryophyllum gastonis-bonnieri*	Quercetin	Flavonoid	Reduces serum glucose in a dose-dependent fashion.	[135]
	Fisetin	Flavonoid	Improves blood glucose homeostasis, lowers methylglyoxal-dependent protein glycation, and mitigates diabetes-related complications.	[136]
	Caffeic acid	Phenol	Reduces hepatic glucose output and enhances adipocyte glucose uptake, insulin secretion, and antioxidant capacity	[113]
	Ferulic acid	Phenol	Improves insulin sensitivity and hepatic glycogenesis, also inhibits gluconeogenesis and maintains insulin signalling to maintain normal glucose homeostasis.	[112]
*Cajanus cajan*	Betulinic acid	Terpenoid	Reduces blood glucose, α-amylase and improves insulin sensitivity as well as pancreas histopathology	[137]
	Pinostrobin	Flavonoid	Reduces the blood sugar level of diabetic mice	[138]
	Genistein	Flavonoid	Inhibits hepatic glucose production, increases β-cell proliferation, reduces β-cell apoptosis, and shows antioxidant activity	[139]
*Costus comosus*	Camphene	Monoterpene	Reduces fasting blood sugar and blood insulin levels.	[140]
	n-Hexadecanoic acid	Fatty acid	α-amylase and α-glucosidase inhibitory and antioxidant activities	[122]
*Costus villosissimus*	Not known		Not known	
*Croton wagneri*	Myrcene	Monoterpene	α-amylase and α-glucosidase inhibitory activities	[141]
*Foeniculum vulgare*	Trans-anethole	Phenylpropanoid	Suppresses diabetic nephropathy in rats by decreasing blood glucose levels and downregulating AT1R and TGF-β1 expressions	[110]
	Fenchone	Monoterpenoid	Protects against increased blood glucose levels and decreased levels of antioxidant enzyme activities in alloxan-induced diabetic rats	[142]
	Estragole	Phenylpropene	α-amylase and lipase inhibitory activity, antioxidant activity	[127]
	Methyl chavicol	Phenylpropene	α-amylase and tyrosinase inhibitory activity, antioxidant activity	[128]
	Limonene	Monoterpene	Inhibits protein glycation, stimulates the uptake of glucose and breakdown of fats, upregulates glucose transporter 1 (GLUT1) expression, and suppresses α-amylase and α-glucosidase	[143]
	α-Phellandrene	Monoterpene	Increases glucose uptake, enhances glycerol-3-phosphate activity and triglyceride accumulation, and regulates adipositic function	[144]
*Glycyrrhiza glabra*	Glycyrrhizin	Saponin	Reduce blood insulin levels and improve tolerance to oral glucose loading and oxidative stress.	[25]
	n-Hexadecanoic acid	Fatty acid	α-amylase and α-glucosidase inhibitory activities and antioxidant activity	[122]
	Dodecanoic acid	Fatty acid	Decreases fasting blood glucose level and induces β-cell regeneration in diabetic rat	[123]
	Carvone	Terpenoid	Improves glycoprotein components and controls glucose metabolism	[145]
*Ilex guayusa*	Chlorogenic acid	Phenol	Increases glucose uptake in L6 muscular cells and raises insulin secretion from the INS-1E insulin-secreting cell line and rat islets of Langerhans.	[111]
	Quercetin-3-O-hexose	Flavonoid	Inhibits the activity of glucose transporter, enhances glucose uptake, reduces hepatic glucose production, protects against pancreatic islet beta-cell, α-glucosidase inhibition	[146]
*Ipomoea carnea*	Spathulenol	Sesquiterpenoid	Strong antioxidant, α-amylase, and α-glucosidase inhibitory activities	[147]
	Caryophyllene oxide	Sesquiterpene	α-amylase and α-glucosidase inhibitory activities and antioxidant activity	[148]
*Juglans neotropica*	Holocellulose	Polysaccharide	Acts as an activator for glucokinase in reducing blood sugar	[131]
	Lignin	Organic polymer	Improves inhibitory effect on α-amylase activity, antioxidant activity	[132]
*Justicia colorata*	No proper evidence found	-	-	-
*Matricaria chamomilla*	Germacrene D	Sesquiterpene	Strong inhibitor of α-glucosidase	[143]
*Myroxylon peruiferum*	Germacrene D	Sesquiterpene	Strong inhibitor of α-glucosidase	[143]
	α-Pinene	Terpene	Inhibition of α-amylase	[149]
	Spathulenol	Sesquiterpenoid	Strong antioxidant, α-amylase, and α-glucosidase inhibitory activities	[147]
	Caryophyllene oxide	Sesquiterpene	α-amylase and α-glucosidase inhibitory activity and antioxidant activity	[148]
	Limonene	Monoterpene	Inhibits protein glycation, stimulates the uptake of glucose and breakdown of fats, upregulates the glucose transporter 1 (GLUT1) expression, and suppresses α-amylase and α-glucosidase	[150]
*Neonelsonia acuminata*	No proper evidence found	-	-	-
*Oreocallis grandiflora*	Myricetin 3-O-β-glucuronide	Flavonoid	Stimulates 2-deoxy-glucose uptake in C2C12 myocytes	[151]
	Isorhamnetin hexuronide	Flavonoid	Decreases glucose level and oxidative stress, modulates lipid metabolism and adipocytic activity	[152]
	Quercetin 3-O-rutinoside	Flavonoid	Regulates whole-body glucose homeostasis; reduces intestinal glucose absorption, insulin secretion, and insulin-sensitizing actions; and enhances glucose utilization in peripheral tissues	[153]
	Quercetin 3-O-β-glucuronide	Flavonoid	Stimulates 2-deoxy-glucose uptake in C2C12 myocytes	[151]
	Isorhamnetin hexoside	Flavonoid	Decreases glucose levels and oxidative stress, modulates lipid metabolism and adipocytic activity	[153]
	Isorhamnetin 3-O-rutinoside	Flavonoid	Decreases glucose level and oxidative stress, modulates lipid metabolism and adipocytic activity	[153]
*Pelargonium graveolens*	Chlorogenic acid	Phenol	Increases glucose uptake in L6 muscular cells and raises insulin secretion from the INS-1E insulin-secreting cell line and rat islets of Langerhans.	[111]
	Quercetin-3-O-hexose	Flavonoid	Inhibits the activity of glucose transporter, enhances glucose uptake, reduces hepatic glucose production, protects against pancreatic islet beta-cell, and inhibits α-glucosidase	[146]
*Persea americana*	β-Caryophyllene	Sesquiterpene	Exhibits selective agonism on cannabinoid receptor type 2 (CB2R), which plays a role in glucose and lipid metabolism, antioxidant, anti-inflammatory activities	[154]
	Caryophyllene oxide	Sesquiterpene	α-amylase and α-glucosidase inhibitory activities and antioxidant activity	[148]
	α-Humulene	Sesquiterpene	Prevents oxidative stress through the reduction mechanism of 8-hydroxy-2-deoksiguanosin in the pancreatic β-cells	[145]
	Catechin	Sesquiterpenoid	Reduces blood sugar source, regulates intestinal functions, improves insulin resistance, and has antioxidant and anti-inflammatory activities	[155]
	Caffeic acid	Phenol	Reduces hepatic glucose output and enhances adipocyte glucose uptake, insulin secretion, and antioxidant capacity	[113]
	Chlorogenic acid	Phenol	Increases glucose uptake in L6 muscular cells and raises insulin secretion from the INS-1E insulin-secreting cell line and rat islets of Langerhans.	[111]
	Coumaric acid	Phenol	Lowers the blood glucose level and gluconeogenic enzymes and increases the activities of hexokinase, glucose-6 phosphatase dehydrogenase, and GSH via increasing levels of insulin.	[97]
	Ferulic acid	Phenol	Improves insulin sensitivity and hepatic glycogenesis, inhibits gluconeogenesis, and maintains insulin signalling to maintain normal glucose homeostasis.	[112]
*Physalis peruviana*	Phytol	Diterpenoid	Stimulates insulin resistance by activation of nuclear receptors and heterodimerization of RXR with PPARγ	[156]
	n-Hexacosane	Alkane	Improves blood glucose, glucose tolerance, glycated hemoglobin, and liver glycogen	[129]
*Piper crassinervium*	Germacrene D	Sesquiterpene	Strong inhibitor of α-glucosidase	[143]
	β-Caryophyllene	Sesquiterpene	Exhibits selective agonism on cannabinoid receptor type 2 (CB2R), which plays a role in glucose and lipid metabolism, antioxidant, and anti-inflammatory activities.	[154]
	Spathulenol	Sesquiterpenoid	Strong antioxidant, α-amylase, and α-glucosidase inhibitory activities	[147]
*Pithecellobium excelsum*	No proper evidence found	-	-	-
*Ruta graveolens*	No proper evidence found	-	-	-
*Siparuna eggersii*	Germacrene D	Sesquiterpene	Strong inhibitor of α-glucosidase	[143]
	Caryophyllene oxide	Sesquiterpene	α-amylase and α-glucosidase inhibitory activities and antioxidant activity	[148]
*Urtica dioica*	Carvacrol	Monoterpenoid	Improves diabetes-related enzymes, insulin resistance, insulin sensitivity, glucose uptake, and antioxidant and anti-inflammatory activities.	[157]
	Carvone	Monoterpenoid	Improves glycoprotein components and controls glucose metabolism	[158]
	Naphthalene	Polycyclic aromatic hydrocarbon	α-glucosidase inhibitory activity	[133]
	(E)-Anethole	Phenylpropanoid	Suppresses diabetic nephropathy in rats by decreasing blood glucose levels and downregulating AT1R and TGF-β1 expressions.	[126]
	Hexahydrofarnesyl acetone	Sesquiterpene	α-glucosidase inhibition activity	[159]
	(E)-β-Ionone	Sesquiterpenoid	α-amylase and α-glucosidase inhibitory activities and antioxidant activity	[160]
	Phytol	Diterpene	Stimulates insulin resistance by activation of nuclear receptors and heterodimerization of RXR with PPARγ	[156]
*Verbena litoralis*	Chlorogenic acid	Phenol	Increases glucose uptake in L6 muscular cells and raises insulin secretion from the INS-1E insulin-secreting cell line and rat islets of langerhans.	[111]
	Caffeic acid	Phenol	Reduction of hepatic glucose output and enhances adipocyte glucose uptake, insulin secretion, and antioxidant capacity	[113]
	Apigenin	Flavonoid	Facilitates glucose-stimulated insulin secretion and prevents ER stress-mediated β-cell apoptosis in the pancreas	[104]
	p-Coumaric acid	Phenol	Lowers the blood glucose level and gluconeogenic enzymes and increases the activities of hexokinase, glucose-6 phosphatase dehydrogenase, and GSH via increasing the level of insulin.	[97]
	Vanillic acid	Phenol	Reduces hyperglycemia and GSH and increases liver enzymes found in diabetic rats, anti-inflammatory activity	[114]
	Ferulic acid	Phenol	Improves insulin sensitivity and hepatic glycogenesis, also inhibits gluconeogenesis, and maintains insulin signalling to maintain normal glucose homeostasis.	[112]

## Data Availability

All data are contained within the article.

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
