# Peer review of "Exploring Folklore Ecuadorian Medicinal Plants and Their Bioactive Components Focusing on Antidiabetic Potential: An Overview"

_plants, 2024, doi:10.3390/plants13111436_

Round 1

Reviewer 1 Report

Comments and Suggestions for Authors

Ecuador has a rich cultural history in ethnobotany, and many Ecuadorian medicinal plants play important role in their people's lives. this review focus on the traditional medicinal plants used in Ecuador for treating Diabetes mellitus and their bioactive phytochemicals, which are mainly responsible for their anti-diabetic properties. This manuscript is interesting and well-prepared, it will help the scientific community to recognize the importance of traditional medicinal plants in Ecuador and attract further researches on the chemical basis and their mode of actions in the treatment of Diabetes mellitus. It will be recommended to the journal after minor revision due to these issues described below:

1.       Page 2 lines 87-90, the Grammer of this sentence should be corrected, such as, “has” should be “have”.

2.       language of the manuscript should be checked by English native speaker.

3.       Among methodology section, all key words employed for this review have to be indicated. Moreover, in addition to “bioactive compound”, other key words for phytochemicals should be considered, such as “chemical”, “constituents”, “secondary metabolite”, “phytochemical”, to overview the research progress on this field.

4.       Line 391, “active compound” is meaningless, use “bioactive” will be better.

5.       As for those bioactive compounds of Ecuadorian medicinal plants, as for “Bryophyllum gastonis bonnieri”, “Flavonoids” should be detailed, namely, the names of compounds should be displayed.

6.       Some discussions about the possible pharmacological mechanisms are also suggested.

Comments on the Quality of English Language

Ecuador has a rich cultural history in ethnobotany, and many Ecuadorian medicinal plants play important role in their people's lives. this review focus on the traditional medicinal plants used in Ecuador for treating Diabetes mellitus and their bioactive phytochemicals, which are mainly responsible for their anti-diabetic properties. This manuscript is interesting and well-prepared, it will help the scientific community to recognize the importance of traditional medicinal plants in Ecuador and attract further researches on the chemical basis and their mode of actions in the treatment of Diabetes mellitus. It will be recommended to the journal after minor revision due to these issues described below:

1.       Page 2 lines 87-90, the Grammer of this sentence should be corrected, such as, “has” should be “have”.

2.       language of the manuscript should be checked by English native speaker.

3.       Among methodology section, all key words employed for this review have to be indicated. Moreover, in addition to “bioactive compound”, other key words for phytochemicals should be considered, such as “chemical”, “constituents”, “secondary metabolite”, “phytochemical”, to overview the research progress on this field.

4.       Line 391, “active compound” is meaningless, use “bioactive” will be better.

5.       As for those bioactive compounds of Ecuadorian medicinal plants, as for “Bryophyllum gastonis bonnieri”, “Flavonoids” should be detailed, namely, the names of compounds should be displayed.

6.       Some discussions about the possible pharmacological mechanisms are also suggested.

Author Response

Thank you very much for taking the time to review this manuscript. We believe your suggestions will help us to improve our research article.  Please find the detailed responses below and the corresponding revisions/corrections highlighted/in track changes in the re-submitted files. "Please see the attachment" which is given below.

Reviewer 2 Report

Comments and Suggestions for Authors

Soham Bhattacharya et  all have submitted the manuscript entitled 'Exploring folklore Ecuadorian medicinal plants and their bio- active components focusing on antidiabetic potential: An over- view'. After close evaluation of the paper I recommend revision according to the next points:

1.The abstract is too general. Why readers must pick up the manuscript among hundreds of review articles regarding treatment of diabetes by medicinal plants? Please underline specific aspects of current review.

2. In Section 2: please clarify time-frame for cited articles. Please justify inclusion/exclusion criteria.

3.  Table 1 contain some confusing information. There are several plants which have indication 'Not known' in the column 'Pharmacological activity as anti-diabetic''. Why these plants are mentioned in the table?

4. Many important details are lacking. I would suggests to update Tanle 1 with importants details.

5. Many important details are lacking. I would suggest to update Table 1 with important details. First of al, it would be reasonable to split information about activity of plants to 'activity in vitr'o 'and activity in vivo'. For activity in vitro please provide details about type of plant extract, tested concetrations, model used, effective concetrations (such as IC50 in case of enzymes inhibition, etc). Please indicate a positive control and its effective concentrations.

6. For models n vivo: please provide information about type of plant extract, animals model used, tested doses, doses of positive control, effects onserved (in numbers such as decrease of insuline level, body weight change, etc.).

7. Are there some clinical date confirming anti-diabetic potential of plants? Please describe in separate section with details of trials.

8. Please discuss how unique the plants used for the treatment of diabetes in Ecuadorian medicine comparing with other systems of medicine (i.e. TCM, Kampo. Ayurveda, European, Russian medicine, etc.). Usually, polyherbal miztures are effective for the treatment of diabetes. Please see for example: https://doi.org/10.1155/2016/5749857;   https://doi.org/10.3389/fphar.2021.697411; https://doi.org/10.3389/fphar.2022.821810; https://doi.org/10.11339/jtm.23.185, etc.

9. Table 2 must be updated with the data about content of identified active compounds in certain plant. It will help to understand how close relation between identified compounds and effects.

10. References are not suitable for conclusions.

11. Lines 505-510 - tjhis information is not related to findings of current paper.

12. Conclusion must be focused in finding of current paper.

Comments on the Quality of English Language

Minor language edition is recommended.

Author Response

(The authors gave the same response as above.)

Round 2

Reviewer 2 Report

Comments and Suggestions for Authors

Authors have revised and updated manuscipt. It could be accepted in present form.